# HYPERADAPT:
# Simple High-Rank Adaptation

**Abel Gurung**                                                    *gurung1@purdue.edu*
*Department of Computer Science*
*Purdue University*

**Joseph Campbell**                                                *joecamp@purdue.edu*
*Department of Computer Science*
*Purdue University*

**Reviewed on OpenReview:** *https://openreview.net/forum?id=uhk13aXVxC*

## Abstract

Foundation models excel across diverse tasks, but adapting them to specialized applications often requires fine-tuning, an approach that is memory and compute-intensive. Parameter-efficient fine-tuning (PEFT) methods mitigate this by updating only a small subset of weights. In this paper, we introduce HYPERADAPT, a parameter-efficient fine-tuning method that significantly reduces the number of trainable parameters compared to state-of-the-art methods like LoRA. Specifically, HYPERADAPT adapts a pre-trained weight matrix by applying row- and column-wise scaling through diagonal matrices, thereby inducing a high-rank update while requiring only $n + m$ trainable parameters for an $n \times m$ matrix. Theoretically, we establish an upper bound on the rank of HYPERADAPT's updates, and empirically, we confirm that it consistently induces high-rank transformations across model layers. Experiments on GLUE, arithmetic reasoning, and commonsense reasoning benchmarks with models up to 14B parameters demonstrate that HYPERADAPT matches or nearly matches the performance of full fine-tuning and state-of-the-art PEFT methods while using orders of magnitude fewer trainable parameters.

## 1 Introduction

Large-scale foundation models have demonstrated remarkable capabilities across diverse tasks, including natural language understanding (Devlin et al., 2019; Radford et al., 2019; Brown et al., 2020), mathematical reasoning (Cobbe et al., 2021), and multi-modal learning (Abdin et al., 2024; Qwen et al., 2025). Despite their broad capabilities, real-world applications often necessitate fine-tuning pre-trained models to better align with domain-specific tasks, constraints, or specialized output formats. Full-model fine-tuning, however, is computationally and memory-intensive given the large number of parameters in state-of-the-art models. Parameter-efficient fine-tuning (PEFT) methods address this challenge by updating only a small subset of parameters. A prominent approach, Low-Rank Adaptation (LoRA) (Hu et al., 2022), reduces the number of trainable parameters by constraining the update to be low-rank. However, its effectiveness depends on the rank of the update; increasing the rank of the low-rank matrices improves performance but increases the number of trainable parameters.

In this work, we take an alternative approach to fine-tuning by observing that pre-trained weight matrices already encode many useful directions. Instead of learning a new low-rank subspace, we can fine-tune a model by reweighting the existing directions. We propose HYPERADAPT, a novel parameter-efficient fine-tuning method that applies row- and column-wise diagonal scaling to a pre-trained weight matrix $W_0$, yielding a fine-tuned matrix $W' = AW_0B$ with just $n + m$ trainable parameters for an $n \times m$ weight matrix. The resulting constrained *high-rank* transformation adjusts the model's sensitivity to different input features and

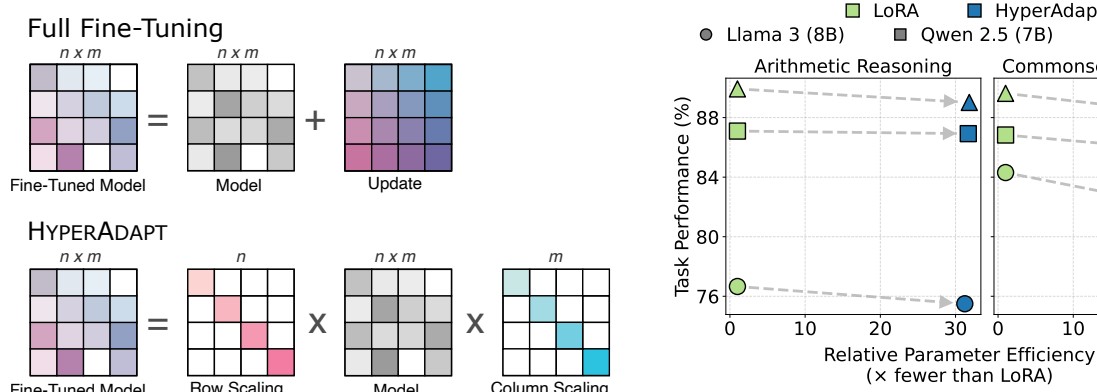

Figure 1: Overview of HYPERADAPT: (Left) Our proposed method, HYPERADAPT, fine-tunes a model by learning row-wise and column-wise diagonal matrices. Unlike full fine-tuning, which requires $n \times m$ trainable parameters, our method yields comparable performance yet only requires $n + m$ trainable parameters. Grayscale values represent frozen parameters, while colored values represent trainable parameters. (Right) Our method achieves similar performance to LoRA across common benchmarks while using up to significantly fewer trainable parameters.

its emphasis on certain output representations, achieving performance comparable to full fine-tuning and state-of-the-art PEFT methods (see Fig. 1).

Our design has three practical benefits. First, it decouples update expressivity from parameter count; by utilizing the model's existing basis, HYPERADAPT achieves effectively high-rank updates without learning new low-rank factors. Second, it is parameter-efficient, yielding up to 34 times fewer parameters than LoRA in our experiments, as simple diagonal scalings suffice to exploit existing pre-trained representations. Third, it adds no additional inference latency, as the scaled weights can be precomputed. Our contributions are:

- **Improved parameter efficiency:** By training diagonal matrices that apply row-wise and column-wise scaling, we significantly reduce the number of trainable parameters compared to prior methods.

- **Competitive performance:** HYPERADAPT achieves model performance comparable to existing PEFT methods such as LoRA, particularly in ultra-low parameter regimes, across widely-used NLP benchmarks.

- **High-Rank adaptation:** We provide a theoretical upper bound on HYPERADAPT's update rank (Lemma 1) and validate it empirically (Sec. 6).

## 2 Preliminaries

**Problem Statement:**

Let $f_\theta$ denote a pre-trained model with parameters $\theta$, mapping an input $x$ to an output $y$. Fine-tuning aims to adapt the model to a downstream task by updating its parameters, producing a new model $f_{\theta'}$ where

$$\theta' = \theta + \Delta\theta$$

Here $\Delta\theta$ is the learned task-specific update. For large models, updating all parameters ($\theta$) is computationally prohibitive. Existing work has demonstrated that fine-tuning large models does not require modifying all parameters. Instead, fine-tuning a small subset of parameters is often sufficient to achieve significant performance improvements on downstream tasks.

The intrinsic dimension hypothesis (Li et al., 2018; Aghajanyan et al., 2020) states that solving a specific task to a desired accuracy generally requires adjusting only a minimal subset of parameters within a low-dimensional subspace of the full parameter space. Building on this principle, LoRA (Hu et al., 2022) extends

the intrinsic dimension hypothesis from the global parameter space to individual weight matrices, showing that low-rank update at this granularity can be sufficient to adapt a pre-trained model. Formally, for a pre-trained weight matrix $W_0 \in \mathbb{R}^{n \times m}$, the goal of PEFT is to efficiently parameterize an update matrix $\Delta W$ such that the adapted weight matrix becomes:

$$W' = W_0 + \Delta W. \tag{1}$$

For LoRA, the update is defined as the product of two low-rank matrices: $\Delta W = BA$ where $B \in \mathbb{R}^{n \times r}$ and $A \in \mathbb{R}^{r \times m}$ for a small rank $r \ll \min(n, m)$. However, the number of trainable parameters scales linearly with the chosen rank $r$, and empirical evidence shows higher ranks generally yield better performance. Thus, achieving stronger adaptation typically involves increasing $r$, which requires more trainable parameters.

## 3 High-Rank Parameter-Efficient Fine-Tuning

It is a well-known phenomenon that over-parameterization of neural networks facilitates easier optimization (Du et al., 2019). Empirical scaling laws(Kaplan et al., 2020; Hoffmann et al., 2022) show that increasing parameters reliably decreases loss, suggesting that larger parameter spaces provide models with more flexibility during training. At the matrix level, allowing updates to affect many orthogonal directions can make adaptation more expressive than constraining updates to a small subspace. Low-rank approaches tailor a handful of directions, whereas high-rank adaptation leverages more directions and can lead to better learning, but is typically memory and compute-intensive.

Pre-trained weight matrices are typically full-rank and already encode many useful directions from the pre-training stage. If we can efficiently reweigh those existing directions, rather than introduce new ones, we can induce high-rank updates with a small parameter budget (see Figure 2). This perspective motivates three principles underlying our approach: (i) exploit directions already present in the pre-trained weight matrix; (ii) combine them efficiently to obtain expressive updates; and (iii) use fewer trainable parameters to reweigh existing directions than learn new ones from scratch.

To efficiently adapt a pre-trained matrix, we propose a constrained high-rank update. Rather than introducing new low-rank factors, we scale the rows and columns of the pre-trained weights to reweigh and recombine its existing directions, yielding a high-rank update with only $n + m$ trainable parameters for an $n \times m$ matrix. This preserves parameter efficiency while exposing many update directions already encoded in the model, improving adaptability without the cost of dense updates. Empirically, our updates

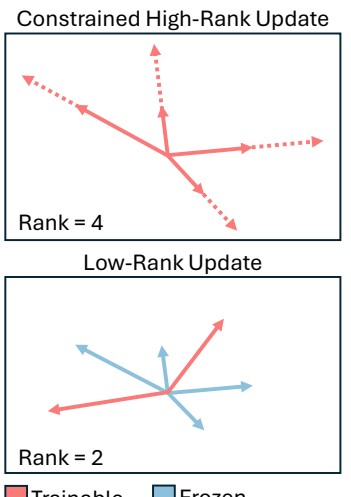

Constrained High-Rank Update

Rank = 4

Low-Rank Update

Rank = 2

■ Trainable   ■ Frozen

Figure 2: HYPERADAPT adjusts a large number of directions via scaling, bootstrapping from pre-trained orthogonal directions (knowledge), achieving a high-rank update. In contrast, low-rank methods modify a limited subset of vectors without any constraint.

realize high normalized rank across most transformer modules (near 1.0 in many layers, see Sec. 6), consistent with the idea that HYPERADAPT exposes many descent directions with a small number of parameters.

### 3.1 HYPERADAPT

In this work, we introduce HYPERADAPT, a parameter-efficient fine-tuning method that achieves high-rank transformations by constraining the form of the update rather than its rank. Given a pre-trained weight matrix $W_0 \in \mathbb{R}^{n \times m}$, we define the fine-tuned update $\Delta W$ to be:

$$\Delta W = AW_0 B - W_0 \tag{2}$$

where $A \in \mathbb{R}^{n \times n}$ and $B \in \mathbb{R}^{m \times m}$ are diagonal matrices. Substituting $\Delta W$ into Eq. 1 yields:

$$\begin{aligned} W' &= W_0 + \Delta W, \\ &= W_0 + A W_0 B - W_0, \\ &= A W_0 B. \end{aligned}$$

The resulting fine-tuned weight matrix $W'$ is then the product of the original weight matrix $W_0$ with two diagonal scaling matrices, $A$ and $B$. Intuitively, $A$ and $B$ matrices scale the row and column of the matrix, selectively stretching or shrinking the latent channels that matter most for the downstream task. This multiplicative reweighting adapts existing representations rather than inventing new structure. Both $A$ and $B$ are initialized to be identity matrices, to ensure that the first forward pass of the model is identical to the original model, so as not to introduce any noise during initialization.

Representing $W'$ using diagonal matrices has two primary benefits: Only $n + m$ trainable parameters are required, and diagonal matrix multiplication can be calculated using element-wise multiplications rather than full matrix multiplications, which is faster. HYPERADAPT is effective despite using only a minimal number of trainable parameters because it produces high-rank updates without explicitly constraining the rank. This enables HYPERADAPT to efficiently adapt pre-trained models to downstream tasks. The rank of $\Delta W$ in equation 2 is upper-bounded by $\min\{2 \cdot \mathrm{rank}(W_0), n, m\}$

> **Lemma 1.** *Let* $W_0 \in \mathbb{R}^{n \times m}$ *and let* $A \in \mathbb{R}^{n \times n}$, $B \in \mathbb{R}^{m \times m}$ *be diagonal matrices. Define* $\Delta W := A W_0 B - W_0$. *Then* $\mathrm{rank}(\Delta W) \leq \min\{2 \cdot \mathrm{rank}(W_0), n, m\}$.

*Proof.* Let $r$ be the $\mathrm{rank}(W_0)$. We know that for any conformable matrix $X$ and $Y$, $\mathrm{rank}(X + Y) \leq \mathrm{rank}(X) + \mathrm{rank}(Y)$. Therefore:

$$\mathrm{rank}(A W_0 B - W_0) \leq \mathrm{rank}(A W_0 B) + \mathrm{rank}(-W_0)$$

Since we define $\Delta W$ to be $A W_0 B - W_0$, we substitute this definition:

$$\mathrm{rank}(\Delta W) \leq \mathrm{rank}(A W_0 B) + \mathrm{rank}(-W_0)$$

Here, $\mathrm{rank}(A W_0 B) \leq \mathrm{rank}(W_0) = r$ because $\mathrm{rank}(XY) \leq \min(\mathrm{rank}\, X, \mathrm{rank}\, Y)$. Therefore, $\mathrm{rank}(A W_0 B) \leq \mathrm{rank}(W_0 B) \leq \mathrm{rank}(W_0)$, and similarly $\mathrm{rank}(A W_0 B) \leq \mathrm{rank}(A W_0) \leq \mathrm{rank}(W_0)$.

$$\mathrm{rank}(\Delta W) \leq r + \mathrm{rank}(-W_0)$$

Additionally, $\mathrm{rank}(-W_0) = \mathrm{rank}(W_0) = r$:

$$\mathrm{rank}(\Delta W) \leq r + r = 2r$$

Furthermore, for any matrix, its rank is always upper-bounded by its dimensions, so the $\mathrm{rank}(\Delta W) \leq \min\{2 \cdot \mathrm{rank}(W_0), n, m\}$. $\qquad\square$

This means that the update matrix induced by HYPERADAPT can achieve a high-rank and is upper-bounded only by the rank of $W_0$. While this transformation cannot increase the rank of $W_0$, as it involves multiplication with diagonal matrices, it nonetheless utilizes the full rank potential of $W_0$ to adapt to the downstream objective. Empirically, we observe this high-rank behavior by examining the singular values of the update produced by HYPERADAPT. In Sec. 6, we report both the rank of the update across all layers of a fine-tuned Qwen-2.5-7B and the spectra of selected fine-tuned weight matrices. Furthermore, similar to prior works, our method introduces **no additional test-time latency** as the modified weights can be precomputed before deployment.

# 4 Related Work

Parameter-efficient fine-tuning is an effective strategy for adapting large pre-trained models to downstream tasks by adjusting only a small subset of model parameters. Early approaches leveraged adapters (Houlsby et al., 2019), lightweight trainable modules which are inserted between transformer layers to enable efficient task-specific adaptation. Other approaches, such as prompt tuning (Lester et al., 2021) and prefix tuning (Li and Liang, 2021), optimize small continuous embeddings at the input or hidden layers to steer model behavior. In contrast, BitFit (Ben Zaken et al., 2022) directly optimizes a small subset of the original model parameters, specifically the bias terms, which results in effective sparse fine-tuning.

**Low-Rank Adaptation and Variants:** Low-Rank Adaptation (LoRA) Hu et al. (2022), discussed in Sec. 2, is one of the most widely adopted methods for fine-tuning large pre-trained models, owing to both its simplicity and flexibility. Many variants have been proposed to further improve performance, including improved initialization strategies (Hayou et al., 2024a; Wang et al., 2024) and asymmetric learning rates (Hayou et al., 2024b). A recent extension, DoRA (Liu et al., 2024), decomposes the pre-trained weight matrix into separate magnitude and direction components, which are then fine-tuned separately.

```python
class LinearHyperAdapt(nn.Module):
    def __init__(self, in_features, out_features, bias=
        None, pretrained_weights=None, train_bias=
        False):
        super().__init__()
        self.weight = nn.Parameter(torch.empty(
            out_features, in_features), requires_grad=
            False)
        self.diag_A = nn.Parameter(torch.ones(
            out_features, 1))
        self.diag_B = nn.Parameter(torch.ones(1,
            in_features))
        if bias is not None:
            self.bias = nn.Parameter(torch.zeros(
                out_features), requires_grad=False)
            self.bias.data.copy_(bias)
        if pretrained_weights is not None:
            self.weight.data.copy_(pretrained_weights)

    def forward(self, x):
        W = self.weight * self.diag_A * self.diag_B
        y = x @ W.T
        if self.bias:
            y = y + self.bias
        return y
```

Listing 1: Torch-style pseudocode of HYPERADAPT linear layer

**High-Rank Adaptation:** In contrast, recent methods such as Singular Vector-guided Fine-Tuning (SVFT) (Lingam et al., 2024) and Vector-based Random Matrix Adaptation (vera) (Kopiczko et al., 2024) aim to induce high-rank updates using a small number of trainable parameters, similar in spirit to HYPERADAPT. SVFT achieves high-rank adaptation by applying singular value decomposition to the pre-trained weight matrix W and fine-tuning only the singular values $\Sigma$, while freezing the singular vectors U and V. However, even though the singular vectors are not updated, they must still be stored in memory which introduces a non-trivial memory overhead. For a weight matrix $W \in \mathbb{R}^{n \times n}$, SVFT must store both $U \in \mathbb{R}^{n \times n}$ and $V \in \mathbb{R}^{n \times n}$ as additional non-trainable parameters, effectively doubling the memory footprint compared to storing W alone.

Similarly, VeRA introduces two large fixed random matrices to project and reconstruct updates, leading to substantial memory consumption. While both SVFT and VeRA achieve high-rank adaptation, their reliance on large auxiliary matrices makes them memory inefficient. HYPERADAPT avoids such overheads as it does not introduce any additional non-trainable parameters.

An alternative approach, Butterfly Orthogonal Fine-Tuning (BoFT)(Liu et al., 2023), introduces a parameter-efficient fine-tuning scheme based on butterfly factorization. This structure allows for the representation of a dense orthogonal matrix as a product of several sparse matrices. However, BoFT trades parameter efficiency for computational efficiency, replacing a single dense matrix operation with a sequence of expensive sparse matrix multiplications in each layer. $IA^3$ (Liu et al., 2022) applies a one-sided diagonal reweighting of the base weight matrix, and scale-and-shift (Lian et al., 2023) can be viewed as $IA^3$ augmented with a trainable bias. For both of these methods, the rank of the update is upper-bounded by $\text{rank}(\Delta W) \leq \text{rank}(W)$, whereas HYPERADAPT's two-sided reweighting yields $\text{rank}(\Delta W) \leq 2 \cdot \text{rank}(W)$. In Sec. 6.2, we also explore the effect of such one-sided transformation.

# 5 Empirical Experiments

To evaluate the effectiveness of our method, we aim to answer two research questions: 1) How does the downstream task performance of models fine-tuned with our method compare to full fine-tuning and existing PEFT methods? 2) How does the number of trainable parameters required by our method compare to those of these other methods?

To answer these questions, we fine-tune four LLMs of varying sizes: RoBERTa-Large (355M) (Liu et al., 2019), Llama-3-8B (et al, 2024), Qwen-2.5-7B (Qwen et al., 2025), and Phi-4 (14B) (Abdin et al., 2024). The fine-tuned models are evaluated on a wide range of NLP tasks spanning four benchmarks: GLUE (Wang et al., 2018), Commonsense Reasoning Benchmark, Arithmetic Reasoning Benchmark and Math Benchmark. Throughout, HYPERADAPT tunes only $n + m$ parameters per $n \times m$ matrix, introducing no inference-time latency because the scaled weights can be precomputed. In our experiments, this yields up to $\approx 34\times$ fewer trainable parameters than LoRA while remaining competitive in accuracy. All experiments were conducted using pre-trained models from the HuggingFace Transformers library (Wolf et al., 2020).

We compare HYPERADAPT against four baselines: four low-rank methods/variants and one high-rank method. The low-rank baselines are: LoRA (Hu et al., 2022); LoRA$_{r=1}$, a rank-1 variant configured such that the number of trainable parameters is the same as in our method; DoRA (weight-decomposed LoRA) (Liu et al., 2024) and DoRA$_{r=1}$ as rank-1 variant for DoRA. For the high-rank baselines, we use VeRA (Vector-based Random Matrix Adaptation).

## 5.1 GLUE Benchmark

We first evaluate our proposed method on the General Language Understanding Evaluation (GLUE) benchmark (Wang et al., 2018) using RoBERTa-Large. The GLUE benchmark (Wang et al., 2018) is a collection of various natural language processing (NLP) tasks designed to evaluate the generalization capabilities of language models. It includes single-sentence classification, sentence-pair classification, and similarity tasks. We primarily use six of its sub-tasks: CoLA (Corpus of Linguistic Acceptability) determines whether a given sentence is grammatically acceptable (Warstadt et al., 2019). SST-2 (Stanford Sentiment Treebank) classifies movie reviews as positive or negative (Socher et al., 2013). MRPC (Microsoft Research Paraphrase Corpus) identifies whether two sentences are semantically equivalent (Dolan and Brockett, 2005). STS-B (Semantic Textual Similarity Benchmark) measures the similarity of two sentences (Cer et al., 2017). QNLI (Question Natural Language Inference) evaluates whether a given passage contains the answer to a question (Rajpurkar et al., 2016). RTE (Recognizing Textual Entailment) is a binary classification task for textual entailment (Dagan et al., 2012). For GLUE, we fine-tune only the Query and Value attention matrices and keep the classifier head frozen, similar to the Hu et al. (2022) setup. To ensure a fair comparison, we use a sequence length of 128 and the same number of training epochs for each task. Full details regarding hyperparameters used can be found in Sec. A.1.

We report our results in Table 1. The results for full fine-tuning and LoRA are taken from Hu et al. (2022). For all methods, we report the average and standard deviation from 5 different seed runs. As shown in Table 1, HYPERADAPT achieves performance comparable to LoRA while using **8 times fewer trainable parameters**. Moreover, it matches the performance of full fine-tuning despite requiring over **1700 times** fewer parameters. Similarly, VeRA also demonstrates strong performance with minimal trainable parameters; however, it introduces 0.5M additional non-trainable parameters during fine-tuning. In contrast, HYPERADAPT achieves 86.0 average with 0.1M trainable parameters without additional non-trainable matrices, while achieving performance comparable to both LoRA and Full fine-tuning.

## 5.2 Arithmetic Reasoning Benchmark

To evaluate the impact of HYPERADAPT on arithmetic reasoning, we follow the experimental setup from Hu et al. (2023). We fine-tune each model on the Math10K dataset(Hu et al., 2023), which comprises training examples from GSM8K(Cobbe et al., 2021) and AQuA (Ling et al., 2017). Models are then evaluated on six downstream arithmetic reasoning tasks. AddSub (Hosseini et al., 2014) tests simple addition and subtraction word problems. SingleEq (Koncel-Kedziorski et al., 2015) involves solving word problems that translate to a

Table 1: GLUE task performance results for RoBERTa-Large. We report Matthew's correlation for CoLA, Pearson correlation for STS-B, and accuracy for other tasks; higher is better. The values for Full FT and LoRA are taken from prior work (Hu et al., 2022). For VeRA, we also report additional non-trainable parameters with red text.

| Method | # Params | SST-2 | MRPC | CoLA | QNLI | RTE | STS-B | QQP | MNLI | Avg. |
|---|---|---|---|---|---|---|---|---|---|---|
| Full FT | 355.0M | 96.4 | 90.9 | 68.0 | 94.7 | 86.6 | 92.4 | 92.2 | 90.2 | 88.9 |
| LoRA | 0.8M | $96.0 \pm 0.3$ | $89.5 \pm 0.2$ | $65.5 \pm 0.8$ | $94.7 \pm 0.3$ | $83.9 \pm 1.6$ | $90.7 \pm 0.4$ | $91.5 \pm 0.1$ | $90.4 \pm 0.1$ | 87.8 |
| LoRA$_{r=1}$ | 0.1M | $96.0 \pm 0.2$ | $85.6 \pm 5.1$ | $62.0 \pm 1.0$ | $94.1 \pm 0.1$ | $77.9 \pm 2.3$ | $84.1 \pm 1.6$ | $90.2 \pm 0.1$ | $89.8 \pm 0.2$ | 85.0 |
| DoRA | 0.8M | $96.0 \pm 0.2$ | $89.3 \pm 0.6$ | $65.8 \pm 0.3$ | $94.6 \pm 0.1$ | $83.5 \pm 1.1$ | $91.0 \pm 0.4$ | $91.6 \pm 0.1$ | $90.4 \pm 0.1$ | 87.8 |
| VeRA | 0.06M | 0.5M | $95.8 \pm 0.3$ | $89.4 \pm 0.5$ | $65.3 \pm 1.5$ | $94.1 \pm 0.2$ | $79.3 \pm 3.4$ | $89.5 \pm 0.8$ | $89.4 \pm 0.2$ | $89.2 \pm 0.2$ | 86.5 |
| **Hyper (Ours)** | **0.1M** | $96.2 \pm 0.2$ | $89.8 \pm 0.3$ | $64.9 \pm 0.7$ | $93.8 \pm 0.1$ | $80.8 \pm 1.3$ | $90.2 \pm 0.3$ | $90.3 \pm 0.1$ | $89.3 \pm 0.1$ | 86.9 |

single algebraic equation. GSM8K (Cobbe et al., 2021) features multi-step grade school problems. AQuA (Ling et al., 2017) focuses on multiple-choice algebra questions, MultiArith(Roy and Roth, 2016) requires sequential arithmetic steps, and SVAMP (Patel et al., 2021) evaluates robustness through perturbed math problems. Because Math10K includes only two of these sub-tasks, this benchmark also assesses generalization to out-of-distribution arithmetic problems.

Table 2: Arithmetic Reasoning results. We report accuracy for all tasks. For all tasks, higher value is better. For VeRA, we also report additional non-trainable parameters with red text.

| Model | Method | # Params (%) | AddSub | SingleEq | GSM8K | AQuA | MultiArith | SVAMP | Avg |
|---|---|---|---|---|---|---|---|---|---|
| **Llama-3-8B** | LoRA$_{r=1}$ | 0.03 | 70.1 | 87.6 | 55.3 | 37.4 | 86.5 | 58.9 | 66.0 |
| | LoRA | 1.03 | 89.6 | 95.9 | 64.4 | 40.9 | 94.2 | 74.9 | 76.7 |
| | DoRA$_{r=1}$ | 0.05 | 71.9 | 85.4 | 54.3 | 36.6 | 86.2 | 59.2 | 65.6 |
| | DoRA | 1.05 | 90.1 | 95.9 | 64.3 | 41.3 | 93.2 | 77.4 | 77.0 |
| | VeRA | 0.02 | 0.37 | 73.7 | 85.6 | 55.0 | 41.3 | 85.0 | 59.5 | 66.7 |
| | **Hyper (Ours)** | **0.03** | 86.3 | 96.7 | 61.9 | 44.1 | 94.2 | 69.8 | 75.5 |
| **Qwen-2.5-7B** | LoRA$_{r=1}$ | 0.03 | 93.4 | 98.4 | 82.6 | 63.4 | 97.5 | 86.5 | 87.0 |
| | LoRA | 1.05 | 92.7 | 98.2 | 79.8 | 70.1 | 98.2 | 83.6 | 87.1 |
| | DoRA$_{r=1}$ | 0.05 | 91.9 | 98.0 | 82.6 | 63.4 | 98.8 | 86.0 | 86.8 |
| | DoRA | 1.07 | 90.9 | 98.6 | 79.7 | 67.7 | 98.7 | 83.4 | 86.5 |
| | VeRA | 0.02 | 0.51 | 94.2 | 98.6 | 82.3 | 66.9 | 98.7 | 88.1 | 88.1 |
| | **Hyper (Ours)** | **0.03** | 92.7 | 98.6 | 79.9 | 68.1 | 98.8 | 83.4 | 86.9 |
| **Phi-4-14B** | LoRA$_{r=1}$ | 0.02 | 93.7 | 98.0 | 86.8 | 68.5 | 98.3 | 87.7 | 88.8 |
| | LoRA | 0.75 | 95.2 | 99.0 | 87.5 | 69.3 | 98.7 | 89.9 | 89.9 |
| | DoRA$_{r=1}$ | 0.04 | 92.9 | 97.8 | 86.8 | 72.0 | 98.7 | 87.7 | 89.3 |
| | DoRA | 0.77 | 95.2 | 99.2 | 87.0 | 69.7 | 98.5 | 89.9 | 89.9 |
| | VeRA | 0.02 | 0.75 | 93.4 | 96.6 | 84.7 | 73.2 | 95.8 | 87.9 | 88.6 |
| | **Hyper (Ours)** | **0.02** | 93.9 | 99.0 | 86.5 | 66.9 | 98.3 | 89.4 | 89.0 |

The results are shown in Table 2. HYPERADAPT demonstrates strong performance relative to existing PEFT methods across all model sizes. Considerably, when compared against LoRA and DoRA, HYPERADAPT shows comparable performance with 34 times fewer parameters for Qwen-2.5-7B and Llama-3-8B and 37 times fewer parameters for Phi-4. LoRA$_{r=1}$ consistently underperforms standard LoRA across all model sizes, highlighting the limitations of aggressive rank reduction. Additionally, LoRA$_{r=1}$ also performs worse than our proposed method, which suggests that HYPERADAPT is able to use the same number of trainable parameters more effectively during fine-tuning.

HYPERADAPT achieves comparable performance with most methods while using fewer parameters. Among the models compared, performance with Llama-3-8B (et al, 2024) stands out. Llama-3-8B is also one of the models released earlier compared to Qwen-2.5-7B (Qwen et al., 2025) and Phi-4 (Abdin et al., 2024), showing that HYPERADAPT is more robust across different models. For VeRA, we report both additional non-trainable parameters and trainable parameters. Both baseline high-rank methods (vera and BoFT) exhibit a significant reduction in trainable parameters compared to low-rank methods. However, HYPERADAPT stands out as more robust than these methods, as demonstrated by the performance in Llama-3-8B, where HYPERADAPT consistently shows **+9%** improvement among these methods.

### 5.3 Commonsense Reasoning Benchmark

For commonsense reasoning benchmarks, we first fine-tuned the models on the Commonsense170K dataset (Hu et al., 2023), an aggregated dataset consisting of eight sub-tasks. Unlike Math10K (10K examples), Commonsense170K offers 170K training instances, allowing us to stress-test HYPERADAPT at a substantially larger data scale and assess its effectiveness. These sub-tasks evaluate a model's ability to reason about everyday scenarios and implicit world knowledge that may not be directly stated in the text. Arc-challenge and Arc-easy (Clark et al., 2018) consist of science exam questions drawn from a variety of sources, Winogrande(Sakaguchi et al., 2019) evaluates pronoun resolution in challenging contexts, SocialInteractionQA (SIQA) (Sap et al., 2019) assesses social and situational reasoning, OpenBookQA (OBQA)(Mihaylov et al., 2018) focuses on science-related multiple-choice questions, BoolQ(Clark et al., 2019) contains yes/no questions from real-world queries, PhysicalInteractionQA(PIQA) (Bisk et al., 2019) tests physical commonsense, and HellaSwag (Zellers et al., 2019) challenges models with grounded commonsense questions.

Table 3: Commonsense Reasoning results. We report accuracy for all tasks. For all tasks, higher value is better. For VeRA, we also report additional non-trainable parameters with red text.

| Model | Method | # Params (%) | ARC-c | ARC-e | WinoGrande | SIQA | OBQA | BoolQ | PIQA | HellaSwag | Avg |
|-------|--------|--------------|-------|-------|------------|------|------|-------|------|-----------|-----|
| **Llama-3-8B** | LoRA$_{r=1}$ | 0.03 | 76.5 | 89.7 | 77.4 | 75.2 | 78.8 | 60.8 | 84.9 | 89.9 | 79.1 |
| | LoRA | 1.03 | 79.4 | 90.3 | 83.0 | 79.8 | 86.0 | 72.5 | 87.9 | 95.5 | 84.3 |
| | DoRA | 1.05 | 79.6 | 90.8 | 83.8 | 80.1 | 84.2 | 73.2 | 87.9 | 95.5 | 84.4 |
| | VeRA | 0.02 \| 0.37 | 74.4 | 89.0 | 74.0 | 73.3 | 78.8 | 61.6 | 84.0 | 84.5 | 77.5 |
| | **Hyper (Ours)** | **0.03** | 78.2 | 89.3 | 79.4 | 76.4 | 80.6 | 67.9 | 86.3 | 92.4 | 81.3 |
| **Qwen-2.5-7B** | LoRA$_{r=1}$ | 0.03 | 88.1 | 95.8 | 76.0 | 78.9 | 87.4 | 70.1 | 88.0 | 92.8 | 84.6 |
| | LoRA | 1.05 | 88.6 | 95.8 | 83.5 | 80.2 | 89.2 | 72.7 | 89.8 | 94.9 | 86.8 |
| | DoRA | 1.07 | 88.5 | 95.9 | 82.4 | 79.8 | 89.6 | 72.8 | 89.6 | 94.6 | 86.7 |
| | VeRA | 0.02 \| 0.51 | 88.0 | 95.4 | 74.8 | 78.6 | 87.8 | 69.2 | 88.4 | 92.6 | 84.3 |
| | **Hyper (Ours)** | **0.03** | 88.3 | 95.3 | 80.1 | 78.8 | 90.4 | 68.6 | 88.7 | 93.7 | 85.5 |
| **Phi-4-14B** | LoRA$_{r=1}$ | 0.02 | 92.8 | 97.9 | 83.5 | 79.6 | 91.6 | 73.4 | 90.5 | 94.0 | 87.9 |
| | LoRA | 0.75 | 93.5 | 98.0 | 87.5 | 81.9 | 93.8 | 74.6 | 92.6 | 95.1 | 89.6 |
| | DoRA | 0.77 | 93.9 | 98.2 | 87.3 | 82.0 | 94.8 | 75.1 | 92.4 | 89.9 | 89.2 |
| | VeRA | 0.02 \| 0.75 | 92.6 | 97.8 | 58.4 | 79.3 | 90.8 | 69.9 | 83.6 | 93.6 | 83.2 |
| | **Hyper (Ours)** | **0.02** | 93.3 | 97.7 | 83.1 | 81.0 | 91.2 | 69.7 | 92.6 | 94.5 | 87.9 |

The results are shown in Table 3. HYPERADAPT remains competitive across all three model families while using dramatically fewer trainable parameters than low-rank baselines. On Llama-3-8B, HYPERADAPT attains an average of 81.3 while training just 0.03% of parameters and also surpasses the high-rank VeRA baseline by +3.8 points on average (81.3 vs. 77.5). We observe the same pattern on Qwen-2.5-7B and Phi-4: HYPERADAPT delivers stronger accuracy to VeRA while maintaining a similar minimal trainable parameter. Relative to LoRA and DoRA, HYPERADAPT is typically within $\approx 2\%$, these trends mirror arithmetic reasoning: HYPERADAPT trades at most a modest accuracy delta for an order-of-magnitude reduction in trained weights.

Notably, HYPERADAPT either matches or exceeds LoRA$_{r=1}$ at the same parameter budget on all three models (Llama-3-8B: +2.2; Qwen-2.5-7B: +0.9; Phi-4-14B: on par), suggesting that our constrained high-rank scaling makes more effective use of the available trainable parameters.

### 5.4 Fine-Tuning With Reasoning Traces

To further evaluate the performance of HYPERADAPT in low-data and long-context settings, we fine-tune Qwen-2.5-7B on the S1 dataset (Muennighoff et al., 2025), which contains $1,000$ high-quality reasoning traces and solutions collected from Gemini's "thinking" model (Google, 2024). Following Muennighoff et al. (2025) setup, we train only on the reasoning traces and solutions, but not the question itself, using the Transformer Reinforcement

Table 4: Performance of fine-tuned reasoning models over math benchmarks. We report accuracy for all tasks (higher is better). For VeRA, we also report additional non-trainable parameters with red text.

| Method | # Params (%) | GSM8K | MATH500 |
|--------|--------------|-------|---------|
| LoRA$_{r=1}$ | 0.03 | 75.7 | 62.6 |
| LoRA | 1.05 | 88.8 | 63.6 |
| DoRA$_{r=1}$ | 0.05 | 74.6 | 61.4 |
| DoRA | 1.07 | 88.9 | 65.0 |
| VeRA | 0.02 \| 0.51 | 80.3 | 60.6 |
| **Hyper (Ours)** | 0.03 | 89.0 | 64.0 |

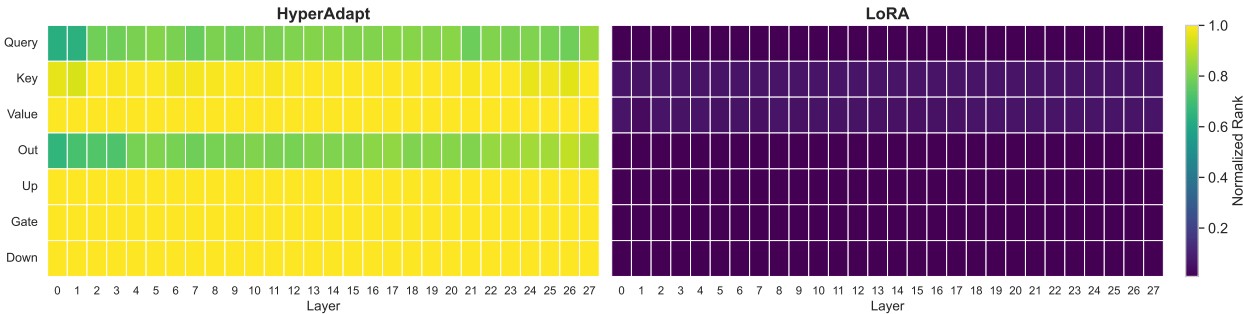

Figure 3: Normalized update rank across all layers of Qwen-2.5-7B after fine-tuning on Commonsense170K. HYPERADAPT produces high-rank updates across most modules effectively utilizing a large fraction of available orthogonal directions.

Learning (TRL) library (von Werra et al., 2020). We set the cut-off length for a given sequence to 16K tokens to assess robustness across longer sequences. The fine-tuned models are evaluated on GSM8K (Cobbe et al., 2021) and MATH500 (Lightman et al., 2023).

As shown in Table 4, HYPERADAPT attains 89.0 on GSM8K and 64.0 on MATH500, effectively matching low-rank baselines with an order of magnitude fewer parameters. Additionally, with the same number of trainable parameters set as HYPERADAPT, $\mathrm{LoRA}_{r=1}$, it substantially underperforms compared to HYPERADAPT, showing that naively shrinking rank is not an adequate substitute for properly fine-tuning models in such constrained trainable parameter settings.

## 6 Rank Analysis and Ablation

To quantify how many orthogonal directions are utilized during fine-tuning, we analyze the empirical rank of the weight update $\Delta \mathrm{W}$. Specifically, given a pre-trained weight $\mathrm{W}_0$ and the fine-tuned counterpart $\mathrm{W}'$, we compute the difference $\Delta \mathrm{W} = \mathrm{W}' - \mathrm{W}_0$ which is the update to the matrix, and examine the singular values of $\Delta \mathrm{W}$. This complements our theoretical upper bound $\mathrm{rank}(\Delta \mathrm{W}) \leq \min\{2 \cdot \mathrm{rank}(\mathrm{W}_0), n, m\}$ (Lemma 1) by quantifying how much of that potential is realized in practice. For each adapted module, we compute the singular value decomposition (SVD) of $\Delta \mathrm{W}$ and count the number of non-trivial singular values. Taking numerical precision into account, we only consider $\left\{ \sigma_i \in \Sigma \mid \sigma_i \geq 1 \times 10^{-2} \right\}$ to be non-trivial. The empirical rank is then normalized by $\mathrm{rank}(\mathrm{W}_0)$, yielding a value in $[0, 1]$ that is comparable across layers of different shapes. Formally,

$$\widehat{r} = \frac{\#\left\{ \sigma_i \in \Sigma \mid \sigma_i \geq 1 \times 10^{-2} \right\}}{\mathrm{rank}(\mathrm{W}_0)} \tag{3}$$

Normalizing by $\mathrm{rank}(\mathrm{W}_0)$ accounts for layer size and also takes into account the rank of the pre-trained matrix, so $\widehat{r}$ reflects how much of the **available subspace** the update actually uses.

We report results on Qwen-2.5-7B fine-tuned on Commonsense170K (Hu et al., 2023). Figure 3 empirically supports our theoretical claim that HYPERADAPT induces high-rank updates. Across layers, HYPERADAPT consistently produces updates with large normalized rank, indicating that most modules exploit a substantial fraction of the available directions. The majority of modules achieve a normalized rank near one, with the exceptions being the Query and Output projection matrices.

For comparison, we also plot the normalized rank for LoRA. Because the pre-trained matrices already have high rank, the normalized rank of LoRA updates collapses toward zero after normalization. This contrast highlights how impactful just $n + m$ trainable parameters can be under HYPERADAPT, yielding updates that are effectively high-rank at the scale of the base matrix.

**Singular Value Trend**. Additionally, we analyze the spectrum of the update matrices, $\Delta \mathrm{W}$, plotting the singular values for the query ($\Delta \mathrm{W}_Q$) and value ($\Delta \mathrm{W}_V$) matrices. Figure 4 shows the spectra for Qwen-2.5-7B and Llama-3-8B fine-tuned on Commonsense170K and Math10K. HYPERADAPT exhibits a slower decay,

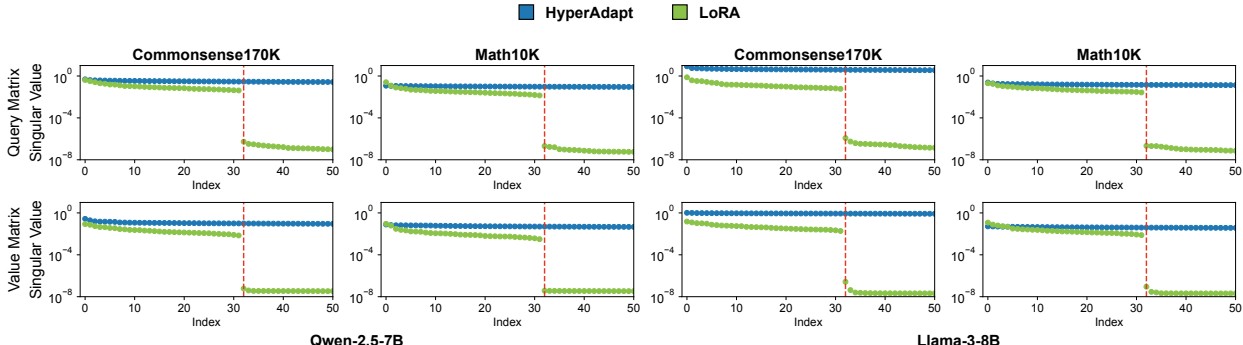

Figure 4: Singular-value spectra of the update matrix $\Delta W$ as given by HYPERADAPT and LoRA for Qwen-2.5-7B and Llama-3-8B. We visualize the first 50 singular values of the update matrix in log scale; values above $1 \times 10^{-2}$ are considered to be non-negligible and contribute to the update's rank. The red dashed line indicates the rank $r$ of LoRA, showing that all values beyond this are negligible. In contrast, HYPERADAPT exhibits a slower decay, reflecting a higher-rank update. The top row corresponds to the Query matrix $\Delta W_Q$ of the 13th layer, and the bottom row corresponds to the Value matrix $\Delta W_V$ of the 13th layer.

indicating a higher-rank update across different weight matrices, models, and datasets. In contrast, LoRA shows a rapid drop in singular values, as expected given its low-rank update.

## 6.1 Learning Rate Sensitivity

To better understand learning rate sensitivity for HYPERADAPT, we performed a sweep over six learning rate using Qwen-2.5-7B: $\{3.0 \times 10^{-2}, 8.0 \times 10^{-3}, 3.0 \times 10^{-3}, 8.0 \times 10^{-4}, 3.0 \times 10^{-4}, 8.0 \times 10^{-5}\}$, trained on Math10K and evaluated on GSM8K. Figure 5 shows that high learning rate ($> 8.0 \times 10^{-3}$) leads to performance degradation, while very low learning ($< 3.0 \times 10^{-4}$) also shows poor performance. We recommend using learning rate in the range $8.0 \times 10^{-4}$ to $3.0 \times 10^{-3}$ for the best performance.

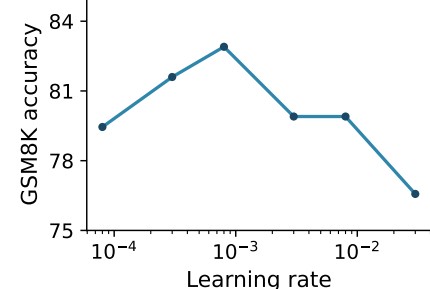

Figure 5: Learning rate sensitivity using Qwen-2.5-7B on GSM8K.

## 6.2 Isolating Effects of Individual Diagonal Matrices

To better understand the expressiveness of an individual diagonal matrix for adaptation, we fine-tune Llama-3-8B on Math10K and follow the same setup as Sec. 5.2. For these experiments, we set either A or B as a trainable and compare against our original results from Sec. 5.2. Table 5 shows that training both diagonal

Table 5: Isolating Effects of individual diagonal matrices on Arithmetic Reasoning Benchmark. We report accuracy for all tasks. For all tasks, higher value is better.

| Method | # Params (%) | AddSub | SingleEq | GSM8K | AQuA | MultiArith | SVAMP | Avg |
|---|---|---|---|---|---|---|---|---|
| Training B only | 0.02 | 80.3 | 92.1 | 58.0 | 38.6 | 90.7 | 64.3 | 70.7 |
| Training A only | 0.02 | 83.0 | 93.3 | 59.0 | 38.6 | 90.3 | 65.5 | 71.6 |
| Training A and B | 0.03 | 86.3 | 96.7 | 61.9 | 44.1 | 94.2 | 69.8 | 75.5 |

matrices consistently improves performance over training either one alone. In particular, training both $A$ and $B$ increases average accuracy from 71.6 (training $A$ only) and 70.7 (training $B$ only) to 75.5 (+3.9 and +4.8 points, respectively). Additionally, these improvements come at nearly identical trainable parameter scales.

## 6.3 Robustness Analysis

To assess the robustness of HYPERADAPT, we trained Roberta-Large with random initialization (scratch) only on the downstream tasks, ensuring the model had no pre-trained knowledge. This would help us better understand how HYPERADAPT will perform in scenarios where the pre-trained model has no relevant pre-training related to the fine-tuning downstream task. In our experimental setup, we trained all linear layers in both the attention and MLP modules, unlike in our fine-tuning experiment, where we only targeted the Query and Value matrices of the attention module. Similar to our fine-tuning setup, the final projection layers are kept frozen. For comparison, we test Full fine-tuning, LoRA, $\text{LoRA}_{r=1}$, and HYPERADAPT. To keep comparisons consistent, even with full fine-tuning, only linear layers are trained, and all other layers are frozen, similar to LoRA and HYPERADAPT. For full fine-tuning, we used the same learning rate as the authors of Roberta(Liu et al., 2019).

Table 6: GLUE task performance results for RoBERTa-Large trained from scratch only on downstream tasks. We report Matthew's correlation for CoLA, Pearson correlation for STS-B, and accuracy for other tasks; higher is better.

| Method | # Params | SST-2 | MRPC | CoLA | QNLI | RTE | STS-B | Avg. |
|---|---|---|---|---|---|---|---|---|
| Full | 302M | 78.3 | 70.6 | 0.0 | 50.5 | 52.7 | 18.6 | 45.1 |
| LoRA | 1.5M | $77.6 \pm 1.0$ | $69.2 \pm 0.6$ | $12.6 \pm 1.9$ | $62.4 \pm 0.4$ | $54.1 \pm 1.2$ | $16.9 \pm 1.4$ | 48.8 |
| $\text{LoRA}_{r=1}$ | 0.4M | $77.1 \pm 1.2$ | $69.7 \pm 0.6$ | $5.9 \pm 2.3$ | $60.6 \pm 0.4$ | $53.4 \pm 0.7$ | $9.8 \pm 0.5$ | 46.1 |
| **Hyper (Ours)** | **0.4M** | $75.3 \pm 0.5$ | $69.2 \pm 0.4$ | $10.3 \pm 5.2$ | $61.1 \pm 0.2$ | $53.1 \pm 0.3$ | $13.8 \pm 0.7$ | 47.1 |

As expected, absolute performance in this scratch regime is substantially lower than in the pre-trained setting Sec. 5.1, since the model must learn linguistic features directly from the supervised task data. However, the goal here is to compare how methods perform relative to each other under an unfavorable initialization. Even in the regime, the diagonal matrices learn to scale the relevant rows and columns, achieving better performance than their LoRA rank-1 counterparts. Also, in this regime, full fine-tuning is comparably worse than the PEFT methods, which is unexpected. We reran full fine-tuning with LoRA's learning rate, too, and found it performs worse, averaging 37.7. This robustness analysis shows that even when the pre-trained model has no relevant knowledge for downstream tasks, HYPERADAPT performs as well as other methods.

## 7 Conclusion

In this work, we introduce HYPERADAPT, a simple yet effective parameter-efficient fine-tuning method that leverages diagonal scaling to achieve high-rank updates, requiring only $n + m$ trainable parameters for an $n \times m$ matrix. We also derived an upper bound on the rank induced by HYPERADAPT, clarifying how the method can express complex changes. Across our evaluations, it matched or nearly matched strong PEFT baselines while training a tiny fraction of the parameters, making it practical when compute or memory are constrained. These results demonstrate that high-rank adaptation can be achieved without the need for expensive auxiliary structures or large ranks, providing a scalable and efficient alternative for adapting foundation models.

**Limitations:** This work focuses exclusively on Transformer-based language models; extending HYPERADAPT to other data domains and models (diffusion models) remains an open direction. HYPERADAPT also assumes that the model is pre-trained, making it an effective tool for adaptation. However, when the model is not pre-trained (random initialization), HYPERADAPT cannot bootstrap and exploit the matrix's representation, which leads to poor learning.

## Acknowledgements

This work used the DeltaAI system at the National Center for Supercomputing Applications through allocation CIS250267 from the Advanced Cyberinfrastructure Coordination Ecosystem: Services and Support (ACCESS)

program, which is supported by National Science Foundation grants 2138259, 2138286, 2138307, 2137603, and 2138296.

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

# A Experiments and Hyperparameters

## A.1 GLUE Benchmark

To keep results consistent, we use the same † setup as Hu et al. (2022) for RoBERTa Large. All of our GLUE experiments have the same max sequence length and epochs for each task. We use the learning rate for DoRA from Hu et al. (2022) since both LoRA and DoRA use similar learning rates.

Table 7: The hyperparameters used for RoBERTa-Large on the GLUE benchmark.

| Method | Dataset | SST-2 | MRPC | CoLA | QNLI | RTE | STS-B | QQP | MNLI |
|---|---|---|---|---|---|---|---|---|---|
| | Optimizer | | | | AdamW | | | | |
| | Warmup Steps | | | | 10 | | | | |
| | LR Schedule | | | | Constant | | | | |
| | Epochs | 10 | 20 | 20 | 10 | 20 | 10 | 20 | 10 |
| | Batch Size | | | | 128 | | | | |
| | Target Layers | | | | Q, V | | | | |
| | Max Seq. Len. | | | | 128 | | | | |
| Hyper | Learning Rate | 3E-03 | 8E-03 | 6E-03 | 3E-03 | 3E-03 | 3E-03 | 4E-03 | 4E-03 |
| VeRA | Learning Rate | 3E-03 | 8E-03 | 6E-03 | 3E-03 | 3E-03 | 3E-03 | 4E-03 | 4E-03 |
| | Rank | | | | 256 | | | | |
| LoRA | Learning Rate | 4E-04 | 3E-04 | 2E-04 | 2E-04 | 4E-04 | 2E-04 | 3E-04 | 3E-04 |
| | Rank | | | | 8 | | | | |
| | LoRA $\alpha$ | | | | 16 | | | | |
| $\text{LoRA}_{r=1}$ | Learning Rate | 4E-04 | 3E-04 | 2E-04 | 2E-04 | 4E-04 | 2E-04 | 3E-04 | 3E-04 |
| | Rank | | | | 1 | | | | |
| | LoRA $\alpha$ | | | | 2 | | | | |
| DoRA | Learning Rate | 4E-04 | 3E-04 | 2E-04 | 2E-04 | 4E-04 | 2E-04 | 3E-04 | 3E-04 |
| | Rank | | | | 8 | | | | |
| | LoRA $\alpha$ | | | | 16 | | | | |

We use the following hyperparameters for fine-tuning for the Arithmetic Reasoning benchmark. The learning rates are based on the suggestions from the original paper.

## A.2 Commonsense Reasoning Benchmark

We use the following hyperparameters for fine-tuning for the Commonsense Reasoning benchmark. The learning rates are based on the suggestions from the original paper.

## A.3 Fine-Tuning With Reasoning Traces

We use the following hyperparameters for fine-tuning for the over reasoning traces. The learning rates are based on the suggestions from the original paper.

## A.4 Decoding Hyperparameters

For commonsense reasoning, we generate at most 32 new tokens. For Arithmetic reasoning, we generate at most 512 new tokens. For math benchmark after fine-tuning on reasoning traces, we generate at most 1024 new tokens.

Table 8: The hyperparameters used for the Arithmetic Reasoning Benchmark.

| Method | Models | Llama-3-8B | Qwen-2.5-7B | Phi-4-14B |
|---|---|---|---|---|
| | Optimizer | | AdamW | |
| | Warmup Steps | | 100 | |
| | Max Grad Norm | | 1.0 | |
| | LR Schedule | | Cosine | |
| | Max Seq. Len | | 512 | |
| | Batch Size | | 256 | |
| | Target Layers | | Q, K, V, O, Gate, Up, Down | |
| | Epochs | | 3 | |
| HYPERADAPT | Learning Rate | | 3e-3 | |
| VeRA | Learning Rate | | 3e-3 | |
| | Rank | 1024 | 1024 | 2048 |
| LoRA$_{r=1}$ | Learning Rate | | 1e-4 | |
| | Rank | | 1 | |
| | LoRA $\alpha$ | | 2 | |
| | LoRA Dropout | | 0.05 | |
| LoRA | Learning Rate | | 1e-4 | |
| | Rank | | 32 | |
| | LoRA $\alpha$ | | 64 | |
| | LoRA Dropout | | 0.05 | |
| DoRA$_{r=1}$ | Learning Rate | | 1e-4 | |
| | Rank | | 1 | |
| | LoRA $\alpha$ | | 2 | |
| | LoRA Dropout | | 0.05 | |
| DoRA | Learning Rate | | 1e-4 | |
| | Rank | | 32 | |
| | LoRA $\alpha$ | | 64 | |
| | LoRA Dropout | | 0.05 | |

Table 9: The hyperparameters used for the Common Sense Reasoning Benchmark.

| Method | Models | Llama-3-8B | Qwen-2.5-7B | Phi-4-14B |
|---|---|---|---|---|
| | Optimizer | | AdamW | |
| | Warmup Steps | | 100 | |
| | Max Grad Norm | | 1.0 | |
| | LR Schedule | | Cosine | |
| | Max Seq. Len | | 256 | |
| | Batch Size | | 256 | |
| | Target Layers | | Q, K, V, O, Gate, Up, Down | |
| | Epochs | | 2 | |
| HYPERADAPT | Learning Rate | | 3e-3 | |
| VeRA | Learning Rate | | 3e-3 | |
| | Rank | 1024 | 1024 | 2048 |
| LoRA$_{r=1}$ | Learning Rate | | 1e-4 | |
| | Rank | | 1 | |
| | LoRA $\alpha$ | | 2 | |
| | LoRA Dropout | | 0.05 | |
| LoRA | Learning Rate | | 1e-4 | |
| | Rank | | 32 | |
| | LoRA $\alpha$ | | 64 | |
| | LoRA Dropout | | 0.05 | |
| DoRA | Learning Rate | | 1e-4 | |
| | Rank | | 32 | |
| | LoRA $\alpha$ | | 64 | |
| | LoRA Dropout | | 0.05 | |

Table 10: The hyperparameters used for the Math Benchmark.

| Method | Models | Qwen-2.5-7B |
|---|---|---|
|  | Optimizer | AdamW |
|  | Warmup Steps | 10 |
|  | Max Grad Norm | 1.0 |
|  | LR Schedule | Cosine |
|  | Max Seq. Len | 16384 |
|  | Batch Size | 64 |
|  | Target Layers | Q, K, V, O, Gate, Up, Down |
|  | Epochs | 5 |
| HYPERADAPT | Learning Rate | 3e-3 |
| VeRA | Learning Rate | 3e-3 |
|  | Rank | 1024 |
| LoRA$_{r=1}$ | Learning Rate | 1e-4 |
|  | Rank | 1 |
|  | LoRA $\alpha$ | 2 |
|  | LoRA Dropout | 0.05 |
| LoRA | Learning Rate | 1e-4 |
|  | Rank | 32 |
|  | LoRA $\alpha$ | 64 |
|  | LoRA Dropout | 0.05 |
| DoRA$_{r=1}$ | Learning Rate | 1e-4 |
|  | Rank | 1 |
|  | LoRA $\alpha$ | 2 |
|  | LoRA Dropout | 0.05 |
| DoRA | Learning Rate | 1e-4 |
|  | Rank | 32 |
|  | LoRA $\alpha$ | 64 |
|  | LoRA Dropout | 0.05 |

Table 11: Decoding hyperparameters used for text generation.

| Parameter | Value |
|---|---|
| Temperature | 0.05 |
| Top-$p$ | 0.40 |
| Top-$k$ | 40 |

