# OpenReview forum: "HyperAdapt: Simple High-Rank Adaptation"
_TMLR — Accepted by TMLR_

### Review · Reviewer_eG3f · 2025-11-24

**Summary Of Contributions:**

This paper introduces HyperAdapt, a novel Parameter-Efficient Fine-Tuning (PEFT) method for adapting large pre-trained models. The core idea is to adapt a pre-trained weight matrix W by applying row-wise and column-wise scaling​ via diagonal matrices A and B, resulting in the adapted weight W' = AW_0B. This formulation requires only n+m trainable parameters per matrix, which is significantly fewer than methods like LoRA (Low-Rank Adaptation). The authors conduct experiments on GLUE, arithmetic reasoning and commonsense reasoning benchmarks using models up to 14B parameters.

**Audience:**

Yes

**Audience Explanation:**

PEFT (parameter-efficient fine-tuning) is an important and interesting topic in large language models. In many real applications, especially in scenarios where computing resources are scarce, PEFT plays an important role.

**Claims And Evidence:**

Yes

**Claims Explanation:**

To demonstrate the effectiveness of the proposed HyperAdapt, the authors conduct experiments on three types of benchmark datasets:
- the general GLUE benchmark, which involves multiple NLU (natural language understanding) tasks
- the mainstream arithmetic reasoning benchmark datasets.
- the commonsense reasoning benchmark datasets.

Though some analyses and discussions are straightforward and superficial, these benchmarks are comprehensive.

The authors also conduct ablation study and parameter sensitivity study to provide a deeper understanding of the proposed method.

**Requested Changes:**

While I acknowledge the comprehensiveness of the conducted benchmark datasets, they still have weaknesses in the empirical experiments:

- Some important implementation details are missing or have less discussion: For example, the authors mention "To ensure a fair comparison, we use a sequence length of 128 and the same number of training epochs for each task", but how many epochs? and why 128 length?
- Explanation of performance drops in CoLA and RTE. In Table 1, we can see that the performance significantly drops on CoLA and RTE (compared with the Full FT). For example, on RTE, the performance drops from 86.6 to 80.8. However, the authors do not provide any discussion on these.
- Involve MNLI in GLUE benchmark: The authors argue that "we exclude QQP and MNLI due to the high computational cost". Given the evidence that MNLI and RTE are NLI tasks, we see the performance drops on RTE (86.6->80.8), thus, I think MNLI is necessary to verify whether the proposed method is suitable for NLI task.
- The improvements brought by HyperAdapt are mixed with different backbones. In Table 2, using llama-3-8, HyperAdapt outperforms LoRA (r=1) by 9.5 points; However, using Qwen-2.5-7B, HyperAdapt underperforms LoRA (r=1). Therefore, I think the performance is highly dependent on the choice of the backbone LLMs. Though the authors discuss that "it may have less data contamination", it is not be verified. I suggest the authors use Qwen-7B (v1 version) to conduct the experiments to verify this claim.
- I also have a minor suggestion: why does Table 1 not have LoRA (r=1) baseline, and Table 2 does not have Full FT baseline?

---

> ### Author Response · Authors · 2025-12-27
>
> 1) Some important implementation details are missing (Hyperparameters)
>
> Please check the appendix (Table A.1) for details on epochs, as they vary across tasks. We will modify the text to make this more straightforward.
>
> 2) Explanation of performance drops in CoLA and RTE, and also a recommendation to include QQP and MNLI.
> We agree there is a performance drop in CoLA and RTE for HyperAdapt compared to Full FT and LoRA. One possible factor could be that we used a batch size of 128 instead of the LoRA paper’s batch size of 4. We primarily chose to do this for practical reasons since a batch size of 4 on modern GPUs would take hours to run any given GLUE task, whereas a batch size of 128 would require only minutes for some GLUE tasks. However, a larger batch size can also mean fewer optimization steps, since the number of epochs is the same across all experiments.
> To ensure a fair comparison, we reran LoRA with a batch size of 128 and, based on your recommendation, we also ran QQP and MNLI across all our GLUE benchmarks.
>
> | Method | # Params | SST-2 | MRPC | CoLA | QNLI | RTE | STS-B | QQP | MNLI | Avg. |
> |---|---:|---:|---:|---:|---:|---:|---:|---:|---:|---:|
> | LoRA | 0.8M | 96.0±0.3 | 89.5±0.2 | 65.5±0.8 | 94.7±0.3 | 83.9±1.6 | 90.7±0.4 | 91.5±0.1 | 90.4±0.1 | 87.8 |
> | DoRA | 0.8M | 96.0±0.2 | 89.3±0.6 | 65.8±0.3 | 94.6±0.1 | 83.5±1.1 | 91.0±0.4 | 91.6±0.1 | 90.4±0.1 | 87.8 |
> | VeRA | 0.06M (+0.5M non-trainable) | 95.8±0.3 | 89.4±0.5 | 65.3±1.5 | 94.1±0.2 | 79.3±3.4 | 89.5±0.8 | 89.4±0.2 | 89.2±0.2 | 86.5 |
> | **HyperAdapt (Ours)** | **0.1M** | **96.2±0.2** | **89.8±0.3** | **64.9±0.7** | **93.8±0.1** | **80.8±1.3** | **90.2±0.3** | **90.3±0.1** | **89.3±0.1** | **86.9** |
>
> With these experiments, we see that smaller batch sizes (as in the results from prior papers) have some effect on specific tasks like CoLA and RTE; however, in both settings (smaller and larger batch sizes), most tasks are stable with regard to batch size. The additional experiments with QQP and MNLI also show that HyperAdapt has a 1% performance difference with 8 times the trainable parameter reduction.
>
>
> 3) The claim about model data contamination affecting performance is unfounded.
>
> Upon further consideration, we agree that running only Llama-3-8B doesn’t fully support the claim that the model is not data-contaminated, so we have decided to remove it and instead state that our model is robust across models.
>
> 4) Missing LoRA (r=1) in GLUE benchmark and questions regarding Full FT in other benchmarks as baseline.
>
> Thank you for this suggestion. Based on your recommendation, we ran LoRA rank=1 in our GLUE benchmark, and here are the results.
>
>
> | Method | # Params | SST-2 | MRPC | CoLA | QNLI | RTE | STS-B | QQP | MNLI | Avg. |
> |---|---:|---:|---:|---:|---:|---:|---:|---:|---:|---:|
> | LoRA (r=1) | 0.1M | 96.0±0.2 | 85.6±5.1 | 62.0±1.0 | 94.1±0.1 | 77.9±2.3 | 84.1±1.6 | 90.2±0.1 | 89.8±0.2 | 85.0 |
> | **HyperAdapt (Ours)** | **0.1M** | **96.2±0.2** | **89.8±0.3** | **64.9±0.7** | **93.8±0.1** | **80.8±1.3** | **90.2±0.3** | **90.3±0.1** | **89.3±0.1** | **86.9** |
>
> Both methods have the same trainable parameters, and we see that HyperAdapt performs slightly better.
>
> Full FT for the autoregressive models is not included as a baseline because these models are big, and we don't have the computational resources to run them.
>
> Once again thank you for taking the time to read our paper and recommending these revisions, it has definitely helped us make this paper better.

---

### Review · Reviewer_7GPn · 2025-11-24

**Summary Of Contributions:**

The paper proposes HyperAdapt, a PEFT method that learns diagonal row and column scaling on pretrained weight, as it involves only $O(n+m)$ trainable parameters per layer. The authors argue that this yields constrained but high-rank updates that can be merged to weight (similarly to LoRA) to introduce no cost at inference time. They report experiments on GLUE with RoBERTa-Large and on several reasoning benchmarks with more recent open-weight models where HyperAdapt performs seemingly well under the regime of ultra-low trainable parameters.

**Audience:**

Yes

**Audience Explanation:**

Effectiveness and limits of parameter-efficient fine-tuning (PEFT) are of high interest to the LLM research community. The finding of effectiveness of high-rank scaling in certain parameter count regime still provides valuable insight for resource-constrained training.

**Claims And Evidence:**

No

**Claims Explanation:**

The authors claim the method "matches / nearly matches" the performance of full fine-tuning and state-of-the-art PEFT methods (LoRA). The provided evidence (Table 1 & 3) shows notable gaps (e.g., -1.8% on GLUE, -3.0% on Llama-3) in several key comparisons. The claims need to be scoped specifically to the ultra-low trainable parameter regime to be accurate. It is reasonable to infer that for tasks requiring the injection of knowledge rarely seen during pre-training, performance will be constrained by the limited number of trainable parameters, even though diagonal scaling achieves high-rank updates. The authors should consider evaluating on a broader range of tasks to verify whether the method remains effective or retains a distinct advantage in such corresponding scenarios.

**Requested Changes:**

1. Please clarify the relationship between HyperAdapt’s core algorithm to other scale/shift-based PEFT methods[1, 2]. These methods also rely on diagonal scalings of existing channels, can be merged back into the weights, and in practice induce high-rank updates with very small parameter counts. At the moment, they are not properly referenced, and the paper’s narrative around “simple high-rank PEFT via diagonal row/column scaling” gives the impression of a new conceptual direction when it is in fact quite close to existing work.

2. It is recommended to put a clear focus on the ultra-low parameter regime. The method’s strength is that it stays competitive when the trainable-parameter budget is pushed down to $O(n+m)$. By design, it is closely tied to this extremely tight parameter budget, and intuitively the adapted model’s performance can be limited by the scaling-only parametrization, so it is important to acknowledge and discuss these potential limitations thoroughly and to add more failure-mode analysis from experiments.


> [1] Liu, Haokun, et al. "Few-shot parameter-efficient fine-tuning is better and cheaper than in-context learning." Advances in Neural Information Processing Systems 35 (2022): 1950-1965.
>
> [2] Lian, Dongze, et al. "Scaling & shifting your features: A new baseline for efficient model tuning." Advances in Neural Information Processing Systems 35 (2022): 109-123.

---

> ### Author Response · Authors · 2025-12-27
>
> 1) Other scale/shift-based PEFT methods
>
> Thank you for pointing this out. We will include these methods in our related work section. IA^3 acts by applying only one diagonal matrix to transform the base matrix, and scale and shift can be seen as IA^3 with additional trainable bias. For both of these methods, the rank of the update is upper-bounded by rank(W), whereas for HyperAdapt, the rank of the update is upper- bounded by 2*rank(W).
>
> Applying a single diagonal matrix transforms the base matrix to $W' = AW$, where $A$ is the diagonal matrix. Using the definition of the update matrix, we get
> $$
> \Delta W = AW - W = (A - I)W.
> $$
> Therefore,
> $$
> \{rank}(\Delta W) = \{rank}((A - I)W) \le \min\big(\{rank}(A - I), \{rank}(W)\big).
> $$
> Since matrix multiplication cannot increase rank, we have $\{rank}(\Delta W) \le \{rank}(W)$.
>
> Additionally, to test how well training only one diagonal matrix would perform, we fine-tuned Llama-3-8B on Math10K and trained A and B individually.
>
> | Method               | # Params (%) | AddSub | SingleEq | GSM8K | AQuA | MultiArith | SVAMP |  Avg |
> | -------------------- | -----------: | -----: | -------: | ----: | ---: | ---------: | ----: | ---: |
> | Training B only      |         0.02 |   80.3 |     92.1 |  58.0 | 38.6 |       90.7 |  64.3 | 70.7 |
> | Training A only      |         0.02 |   83.0 |     93.3 |  59.0 | 38.6 |       90.3 |  65.5 | 71.6 |
> | **Training A and B** |         0.03 |   86.3 |     96.7 |  61.9 | 44.1 |       94.2 |  69.8 | 75.5 |
>
> From this experiment, we see that having both diagonal matrices is more beneficial with negligible additional trainable parameters.
>
>
> 2) Recommended to put a clear focus on the ultra-low parameter regime.
>
> We will update our text to highlight our focus and benefit on “ultra” low trainable parameter regime”
>
>
> 3) Failure-mode analysis and “tasks requiring the injection of knowledge rarely seen during pre-training”
>
> Although our limitations section stated that HyperAdapt can only adapt to tasks similar to the pre-training tasks, we agree that including a failure-mode analysis experiment would help us better understand this claim. To answer both failure mode analysis and the limitations of knowledge injection, we performed the following experiment: We trained Roberta-Large with random initialization (scratch) only on the downstream tasks, so that this model would have no pre-trained knowledge. In our experimental setup, we trained all linear layers in both the attention and MLP modules, unlike in our fine-tuning experiment, where we only targeted the Query and Value matrices of the attention module. Similar to our fine-tuning setup, the final projection layers are kept frozen. For comparison, we test Full fine-tuning, LoRA, LoRA_r=1, and HyperAdapt. To keep comparisons consistent, even with full fine-tuning, only linear layers are trained, and all other layers are frozen, similar to LoRA and HyperAdapt. For full fine-tuning, we copied the same learning rate as the authors of Roberta [1].
> | Method                | # Params |    SST-2 |     MRPC |     CoLA |     QNLI |      RTE |    STS-B | Avg. |
> | --------------------- | -------: | -------: | -------: | -------: | -------: | -------: | -------: | ---: |
> | Full                  |     302M |     78.3 |     70.6 |      0.0 |     50.5 |     52.7 |     18.6 | 45.1 |
> | LoRA                  |     1.5M | 77.6±1.0 | 69.2±0.6 | 12.6±1.9 | 62.4±0.4 | 54.1±1.2 | 16.9±1.4 | 48.8 |
> | LoRA$_{r=1}$          |     0.4M | 77.1±1.2 | 69.7±0.6 |  5.9±2.3 | 60.6±0.4 | 53.4±0.7 |  9.8±0.5 | 46.1 |
> | **HyperAdapt (Ours)** | **0.4M** | 75.3±0.5 | 69.2±0.4 | 10.3±5.2 | 61.1±0.2 | 53.1±0.3 | 13.8±0.7 | 47.1 |
>
> Initially, we expected HyperAdapt to perform worse compared to LoRA_r=1 in this setting since we believed that having a one-rank structure would be more beneficial than just scaling rows and columns of a random matrix. However, to our surprise, the diagonal matrices learns to scale the relevant rows and columns to learn more than their LoRA rank-1 counterpart. Also, in this regime, full fine-tuning is comparably worse than the PEFT methods, which is unexpected. We reran full fine-tuning with LoRA's learning rate, too, and found that it performs worse, with an average of 37.7. This failure mode analysis highlights that even under conditions where the pre-trained model has zero relevant knowledge related to downstream tasks, HyperAdapt adequately performs as well as other methods.
>
> [1] https://arxiv.org/abs/1907.11692
>
> Once again thank you for taking the time to read our paper and recommending these revisions, it has definitely helped us make this paper better.

---

### Review · Reviewer_rPAv · 2025-12-12

**Summary Of Contributions:**

This paper proposes a new PEFT method that is high rank. This is achieved by multiplying 1D weights on rows and columns of the 2D weights matrices.

Strength:
1. The method vastly reduces the number of trainable parameters from N*M to N+M.
2. Good resulss are obtained on multiple LLMs and multiple benchmark tasks.
3. The method is simple to implement and well presented in the paper.

Weakness:
1. This method is almost a special case of Dora (e.g. rank=1), where only the magnitude is trained but not many of the directions. Especially when the network includes two consecutive layers, the scaling in the previous layer output would almost cancel out with the scaling of the next layer input, making HyperAdapt more similar to Dora.
2. Empirical results are not compared with Dora (rank=1), although Lora (rank=1) is included. This weakens the claim that “Hyper is 34x more parameter efficient than Dora”.
3. Hyper supports only the extremely low parameter regime and is evaluated on easy benchmarks. It is not clear that the method works well enough on challenging tasks. It is not clear either whether this extreme efficient regime achieves a good performance-storage trade-off — the results are a bit worse than other methods like Dora.

**Audience:**

Yes

**Audience Explanation:**

It may be possible that some individuals, who focus on a really extreme PEFT regime, might be interested in the findings, but it is not clear that this extreme regime is an empirically good operating point for LLMs.

In addition, for these individuals, using Dora (rank=1) might achieve better results with a similar number of trained parameters.

**Claims And Evidence:**

No

**Claims Explanation:**

The main concern is the claim that "Hyper is 34x more parameter efficient than Dora". This was compared with the standard Dora that aims for challenging tasks. Hyper is not compared with a Dora that is rank=1. This paper does not include more challenging tasks where results are below 60.

**Requested Changes:**

Including Dora (rank = 1) in the tables would help to understand the contribution over the literature.

One interesting study could be to study row only for the whole network, and column only, compared with row + column, and see how they are complementary to each other.

---

> ### Author Response · Authors · 2025-12-27
>
> 1) Differences compared to DoRA and experiments with DoRA rank=1.
>
> We agree that DoRA has multiplicative factors, but even with rank=1, it also has an additive factor (a low-rank matrix of rank-1). So even in its rank=1 case, HyperAdapt cannot be seen as a special case of DoRA. This can be quickly verified by comparing the number of trainable parameters in DoRA’s rank one case; for an N by N matrix, it would need 3N number of trainable parameters, whereas HyperAdapt requires 2N number of trainable parameters.
> We agree with your recommendation for adding DoRA’s rank=1 case for comparison to understand the contribution, and here are the results: we fine-tuned DoRA rank=1 on Math10K just as our other methods for the Arithmetic Reasoning Benchmark.
>
> | Model           | Method           | # Params (%) | AddSub | SingleEq | GSM8K | AQuA | MultiArith | SVAMP |  Avg |
> | --------------- | ---------------- | -----------: | -----: | -------: | ----: | ---: | ---------: | ----: | ---: |
> | **Llama-3-8B**  | DoRA$_{r=1}$     |         0.05 |   71.9 |     85.4 |  54.3 | 36.6 |       86.2 |  59.2 | 65.6 |
> |                 | **Hyper (Ours)** |     **0.03** |   86.3 |     96.7 |  61.9 | 44.1 |       94.2 |  69.8 | 75.5 |
> | **Qwen-2.5-7B** | DoRA$_{r=1}$     |         0.05 |   91.9 |     98.0 |  82.6 | 63.4 |       98.8 |  86.0 | 86.8 |
> |                 | **Hyper (Ours)** |     **0.03** |   92.7 |     98.6 |  79.9 | 68.1 |       98.8 |  83.4 | 86.9 |
> | **Phi-4-14B**   | DoRA$_{r=1}$     |         0.04 |   92.9 |     97.8 |  86.8 | 72.0 |       98.7 |  87.7 | 89.3 |
> |                 | **Hyper (Ours)** |     **0.02** |   93.9 |     99.0 |  86.5 | 66.9 |       98.3 |  89.4 | 89.0 |
>
> Here we see that for Qwen-2.5-7B and Phi-4-14B, the performance of DoRA rank=1 and HyperAdapt are similar, although DoRA rank=1 still requires slightly more trainable parameters than HyperAdapt. Importantly, when comparing DoRA rank=1 and HyperAdapt on Llama-3-8B, we see that HyperAdapt scores 10% higher, showing that HyperAdapt is more robust across different models.
>
> Additionally, we also fine-tuned Qwen-2.5-7B on the S1 dataset with DoRA rank=1 as our section 5.4 setup, where we evaluate on GSM8K and MATH500.
> | Method       | # Params (%) | GSM8K | MATH500 |
> | ------------ | -----------: | ----: | ------: |
> | DoRA$_{r=1}$ |         0.05 |  74.6 |    61.4 |
> | **Hyper**    |         0.03 |  89.0 |    64.0 |
>
>
> We find that DoRA rank=1’s performance is similar to LoRA rank=1’s performance, where the difference in GSM8K is much higher compared to HyperAdapt.
>
> 2) Row-only and Column-only ablation study
>
>
> Thank you for suggesting this interesting ablation study. We fine-tuned Llama-3-8B on the same setup as our section 5.2 experiment. We found that training both A and B, compared to training only A or B, yields better performance while operating on a relatively similar trainable parameter scale.
>
> | Method               | # Params (%) | AddSub | SingleEq | GSM8K | AQuA | MultiArith | SVAMP |  Avg |
> | -------------------- | -----------: | -----: | -------: | ----: | ---: | ---------: | ----: | ---: |
> | Training B only      |         0.02 |   80.3 |     92.1 |  58.0 | 38.6 |       90.7 |  64.3 | 70.7 |
> | Training A only      |         0.02 |   83.0 |     93.3 |  59.0 | 38.6 |       90.3 |  65.5 | 71.6 |
> | **Training A and B** |         0.03 |   86.3 |     96.7 |  61.9 | 44.1 |       94.2 |  69.8 | 75.5 |
>
>
>
> Once again thank you for taking the time to read our paper and recommending these revisions, it has definitely helped us make this paper better.

---

### Author Response · Authors · 2025-12-27
**Revision Updates**

- GLUE Benchmark experiments added: LoRA, LoRA rank-1 with batchsize 128
- GLUE Benchmark experiments added: QQP and MNLI experiments for all methods
- DoRA rank-1 experiments added: Ran DoRA rank-1 for Arithmetic Reasoning Benchmark and also fine-tuned on reasoning traces for sec 5.4
- Robustness Analysis: Trained Roberta-Large from scratch for Full fine-tuning, LoRA, LoRA-rank-1 and HyperAdapt and evaluated on GLUE Benchmark
-Isolating Effects of Individual Diagonal Matrices: Fine-tuned Llama-3-8B on A and B diagonal matrices separately to test the individual effects of each transformation
- All text changes updated are in red.

---

### Decision · Action_Editor_qEGd · 2026-03-27

**Recommendation:** Accept as is

**Additional Comments:**

Only one of the reviewers submitted a final recommendation, which was to accept the paper. After carefully reading the author's answer to the other two reviews, I am confident that the issues raised by those 2 reviewers have been properly addressed in the revision. I thus give the final recommendation to accept this submission.

**Audience:**

Yes

**Audience Explanation:**

Yes, PEFT methods are very relevant to TMLR's audience.

**Claims And Evidence:**

Yes

**Claims Explanation:**

The claims made in the revision are well supported. The initial reviews raised a couple of claims that were not substantially supported. Those were either removed, rephrased, or additional evidence were included for support.